# ATP-dependent modulation of MgtE in Mg$^{2+}$ homeostasis

Atsuhiro Tomita[1], Mingfeng Zhang[2], Fei Jin[3], Wenhui Zhuang[2], Hironori Takeda[4], Tatsuro Maruyama[5], Masanori Osawa[5], Ken-ichi Hashimoto[6], Hisashi Kawasaki[6], Koichi Ito[7], Naoshi Dohmae[8], Ryuichiro Ishitani[1], Ichio Shimada[5], Zhiqiang Yan [2,9], Motoyuki Hattori[3] & Osamu Nureki[1]

Magnesium is an essential ion for numerous physiological processes. MgtE is a Mg$^{2+}$ selective channel involved in the maintenance of intracellular Mg$^{2+}$ homeostasis, whose gating is regulated by intracellular Mg$^{2+}$ levels. Here, we report that ATP binds to MgtE, regulating its Mg$^{2+}$-dependent gating. Crystal structures of MgtE–ATP complex show that ATP binds to the intracellular CBS domain of MgtE. Functional studies support that ATP binding to MgtE enhances the intracellular domain affinity for Mg$^{2+}$ within physiological concentrations of this divalent cation, enabling MgtE to function as an in vivo Mg$^{2+}$ sensor. ATP dissociation from MgtE upregulates Mg$^{2+}$ influx at both high and low intracellular Mg$^{2+}$ concentrations. Using site-directed mutagenesis and structure based-electrophysiological and biochemical analyses, we identify key residues and main structural changes involved in the process. This work provides the molecular basis of ATP-dependent modulation of MgtE in Mg$^{2+}$ homeostasis.

[1] Department of Biological Sciences, Graduate School of Science, The University of Tokyo, 2-11-16 Yayoi, Bunkyo-ku, Tokyo 113-0032, Japan. [2] State Key Laboratory of Medical Neurobiology, Collaborative Innovation Center of Genetics and Development, Institute of Brain Science, Department of Physiology and Biophysics, School of Life Sciences, Fudan University, 2005 Songhu Road, Yangpu District, Shanghai 200438, China. [3] State Key Laboratory of Genetic Engineering, Collaborative Innovation Center of Genetics and Development, Department of Physiology and Biophysics, School of Life Sciences, Fudan University, 2005 Songhu Road, Yangpu District, Shanghai 200438, China. [4] Faculty of Life Science, Kyoto Sangyo University, Kamigamo-motoyama, Kita-ku, Kyoto 603-8555, Japan. [5] Graduate School of Pharmaceutical Sciences, The University of Tokyo, Hongo, Bunkyo-ku, Tokyo 113-0033, Japan. [6] Department of Green and Sustainable Chemistry, Tokyo Denki University, 5 Asahi-cho, Senju, Adachi-ku, Tokyo 120-8551, Japan. [7] Department of Computational Biology and Medical Sciences, Graduate School of Frontier Sciences, The University of Tokyo, 5-1-5 Kashiwanoha, Kashiwa-shi, Chiba 277-8562, Japan. [8] Biomolecular Characterization Unit, RIKEN Center for Sustainable Resource Science, 2-1 Hirosawa, Wako-shi, Saitama 351-0198, Japan. [9] Department of Human Anatomy, School of Basic Medicine Sciences, Southwest Medical University, Luzhou, Sichuan 646000, China. Atsuhiro Tomita, Mingfeng Zhang and Fei Jin contributed equally to this work. Correspondence and requests for materials should be addressed to Z.Y. (email: zqyan@fudan.edu.cn) or to M.H. (email: hattorim@fudan.edu.cn) or to O.N. (email: nureki@bs.s.u-tokyo.ac.jp)

Mg$^{2+}$ is the most abundant divalent cation in living organisms and an essential element for numerous physiological activities, such as catalysis by hundreds of enzymes, cell membrane stabilization, and ATP utilization[1, 2]. Accordingly, abnormalities in Mg$^{2+}$ homeostasis are associated with various diseases, including diabetes, obesity, and cardiovascular disease[3]. Therefore, Mg$^{2+}$ homeostasis is a crucial mechanism for both eukaryotic and prokaryotic species, and Mg$^{2+}$ channels and transporters play a central role in Mg$^{2+}$ homeostasis[4, 5].

MgtE is a widely distributed Mg$^{2+}$ channel in both prokaryotes and eukaryotes[6]. The bacterial MgtE channels are highly selective for Mg$^{2+}$, and are involved in the maintenance of the intracellular Mg$^{2+}$ concentration[7, 8]. Likewise, the eukaryotic homologs of MgtE, the solute carrier 41 (SLC41) family proteins, also permeate Mg$^{2+}$ ions[9, 10] and are implicated in Mg$^{2+}$ homeostasis[10–12]. Several mutations in the *SLC41* genes are related to Parkinson's disease[13], diabetes[14], and nephronophthisis[15].

We previously reported the crystal structures of the full-length *Thermus thermophilus* MgtE (TtMgtE) in the presence of Mg$^{2+}$, its transmembrane domain in the presence of Mg$^{2+}$, and its cytosolic domain in the presence and absence of Mg$^{2+}$ [8, 16, 17]. The full-length structure in the presence of Mg$^{2+}$ revealed that MgtE forms a homodimer, consisting of five transmembrane helices, one plug helix, and the cytosolic region composed of the N-terminal domain and the tandemly repeated CBS domains. A comparative analysis of the Mg$^{2+}$-bound and Mg$^{2+}$-free cytosolic domain structures, together with molecular dynamic simulations[18], revealed that Mg$^{2+}$ binding to the cytosolic domain stabilizes the closed conformation of MgtE, suggesting a Mg$^{2+}$ homeostasis mechanism in which the MgtE cytosolic domain acts as a Mg$^{2+}$ sensor to regulate the Mg$^{2+}$ influx. Consistently, the subsequent electrophysiological analysis demonstrated that intracellular Mg$^{2+}$ binding to the MgtE cytosolic domain inhibited the channel opening of MgtE[8].

However, in the previously reported electrophysiological analysis of MgtE, the threshold of intracellular Mg$^{2+}$ for the channel inactivation was between 5 and 10 mM, which is much higher than the physiological intracellular Mg$^{2+}$ concentration ($\sim 1$ mM)[8, 19]. Therefore, this discrepancy implied the existence of additional regulatory factors for the Mg$^{2+}$-dependent gating of MgtE under physiological conditions.

The CBS domain in the cytosolic region of MgtE possesses regulatory sites for Mg$^{2+}$ [8, 16], and includes a nucleotide binding site to regulate the activity of associated enzymes or transporters, such as the human ClC5 Cl$^-$ transporter, in response to the binding of ATP or other nucleotides[20]. However, it remains unclear whether and how Mg$^{2+}$ channels are modulated by ATP or other nucleotides for Mg$^{2+}$ homeostasis.

In this study, we demonstrate that ATP binds to MgtE and affects the Mg$^{2+}$-dependent gating of MgtE, using isothermal titration calorimetry (ITC) and electrophysiology, respectively. We also determine the crystal structures of the full-length and the cytosolic domain of MgtE in complex with ATP, which reveal that ATP is recognized by the CBS domain of MgtE. Our structure-based electrophysiological, genetic, and biochemical analyses provide functional and structural insights into the ATP-dependent regulation of MgtE gating during Mg$^{2+}$ homeostasis.

## Results

### ATP binds to MgtE
To examine whether ATP and other nucleotides are additional regulatory factors of MgtE, we first measured the binding affinities of ATP, ADP, and GTP to MgtE, using ITC (Fig. 1a). We also measured the binding affinity of ATP in the presence of Mg$^{2+}$ (Fig. 1a). MgtE exhibited a $K_d$ of 415 μM for ATP and a $K_d$ of 763 μM for ADP, while no interaction was detected between MgtE and GTP. MgtE exhibited higher affinity to ATP in the presence of Mg$^{2+}$, with a $K_d$ of 172 μM. These $K_d$ values of MgtE for ATP are comparable to the previously reported $K_d$ values of other CBS domain-containing proteins for ATP[20, 21]. Considering the physiological intracellular concentrations of ATP (1–10 mM) and ADP (0.5–1.5 mM)[22], these results indicated that MgtE would exist primarily in the complex with ATP in vivo, and that ATP would possibly be an additional regulatory factor of MgtE. To further examine the ATP-mediated effect on the Mg$^{2+}$-dependent conformational change of MgtE, we performed protease protection assays with a concentration gradient of Mg$^{2+}$, in the presence and absence of ATP (Fig. 1b). A previous protease protection assay revealed that MgtE was highly susceptible to proteolysis at low-Mg$^{2+}$ concentrations, but not at high-Mg$^{2+}$ concentrations, presumably due to the stabilization of the closed conformation[18]. While we observed the Mg$^{2+}$-dependent protease resistance in both the presence and absence of ATP, intriguingly, we detected additional bands with high $M_r$ in the presence of ATP (Fig. 1b). Furthermore, we observed a similar effect with ADP, but not GTP (Fig. 1c). The N-terminal sequencing of the proteolytic fragments revealed that the additionally protected fragments correspond to the full-length MgtE (Supplementary Fig. 1). Overall, these results indicated that ATP binding modulates the Mg$^{2+}$-dependent conformational stabilization of the closed state of MgtE.

### ATP effect on the MgtE channel gating
To test the effect of the ATP binding on the channel gating by MgtE, we performed the patch clamp analysis of MgtE using giant proteoliposomes under conditions with 0–10 mM [Mg$^{2+}$]$_{in}$, and either 0 or 3 mM ATP (Fig. 2). As the [Mg$^{2+}$]$_{in}$ in the absence of ATP in the bath solution increased from 0.2 to 10 mM, the open probability decreased (Fig. 2a, d). While the channel was still active in the presence of 5 mM Mg$^{2+}$ in the bath solution, almost no current was detected in the presence of 10 mM Mg$^{2+}$ (Fig. 2a, d). These results are consistent with the previous patch clamp analysis of MgtE[8], showing the intracellular Mg$^{2+}$-inhibition of the channel gating (Fig. 2a, d). Likewise, under the 3 mM ATP conditions, the open probability decreased as the [Mg$^{2+}$]$_{in}$ in the bath solution increased from 0.2 to 10 mM (Fig. 2b, d). However, in the presence of 3 mM ATP in the bath solution, the channel was completely inactivated at 3 mM [Mg$^{2+}$]$_{in}$, unlike the MgtE under the ATP-free conditions (Fig. 2a). Therefore, these results suggested that intracellular ATP binding to MgtE enhances the affinity of MgtE for intracellular Mg$^{2+}$ within the physiological range ($\sim 1$ mM)[19], which would allow the MgtE cytosolic domain to act as a Mg$^{2+}$ sensor in vivo.

In addition, our results demonstrated that the open probabilities of MgtE in the absence of ATP in the bath solution were higher at both high- and low-intracellular Mg$^{2+}$ concentrations, as compared to those in the presence of 3 mM ATP (Fig. 2d), showing that the ATP dissociation promotes the Mg$^{2+}$ influx into the cells.

### Structure of the MgtE–ATP complex
To gain further insights into the ATP-dependent modulation of the channel gating in MgtE, we determined the crystal structure of the full-length MgtE in complex with ATP at 3.6 Å resolution (Fig. 3a, b, Supplementary Figs. 2a, b and 3a and Table 1). The overall structure represents the closed conformation of the channel, and is essentially identical to the previously reported full-length structure, with a root mean squared deviation (RMSD) value of 0.38 Å for all Cα atoms (Fig. 3a). The electron density for Mg$^{2+}$ corresponds to the Mg$^{2+}$-binding sites (Mg1–Mg7) in the

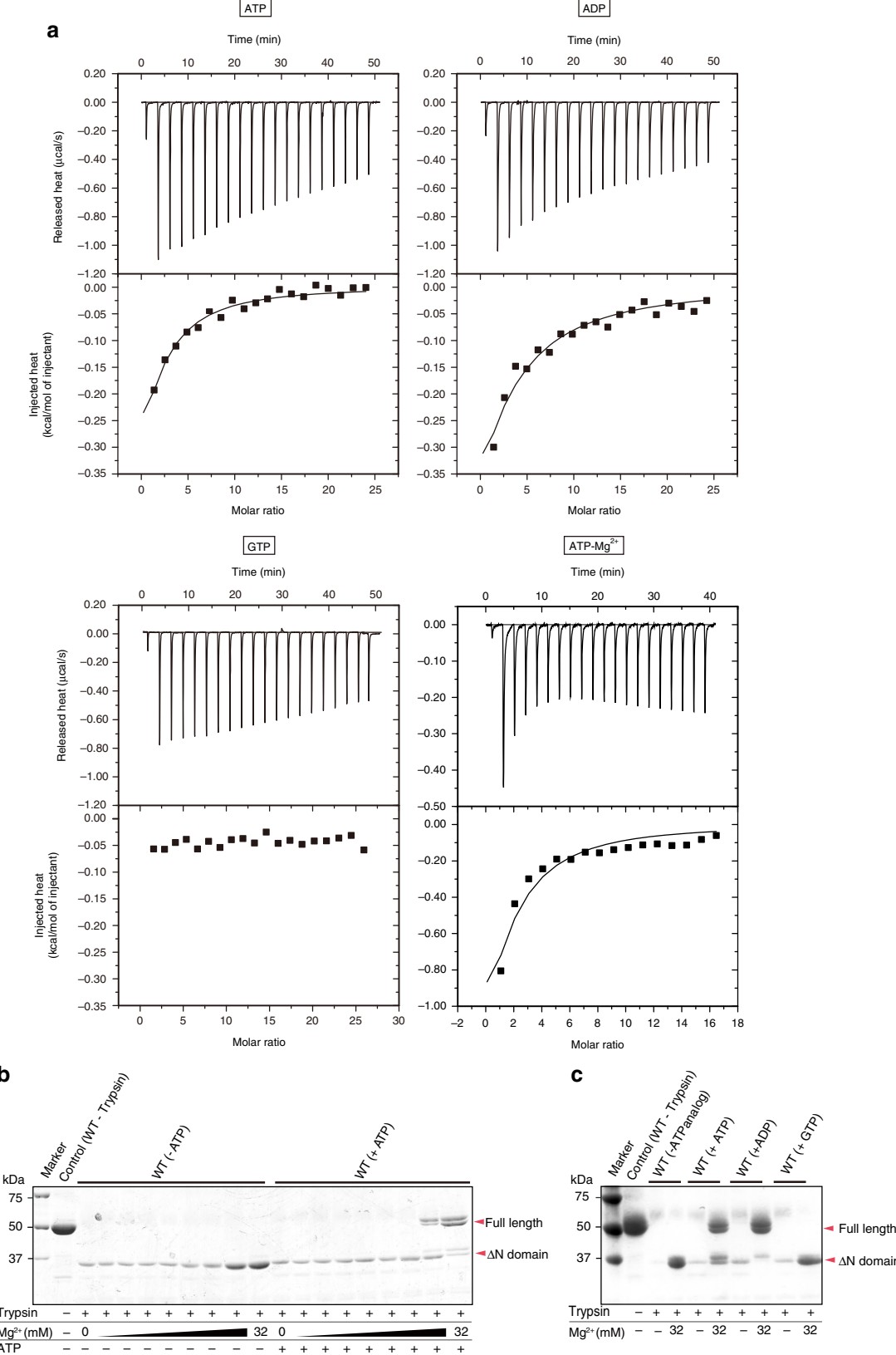

**Fig. 1** Effects of ATP and other nucleotides on MgtE. **a** Isothermal titration calorimetry (ITC) data of MgtE with ATP and other nucleotides. The data were obtained from MgtE with ATP, ADP, GTP, and ATP+Mg$^{2+}$. The raw ITC data and the plots of injected heat for 20 automatic injections of 10 mM nucleotide solution into the sample cell containing the MgtE solution are shown. Measurements were repeated twice, and similar results were obtained. **b, c** Protease protection of MgtE by Mg$^{2+}$ and nucleotides. The numbers on the left side indicate the molecular masses (in kilodaltons) of the markers. The details of the cleavage sites are described in Supplementary Fig. 1

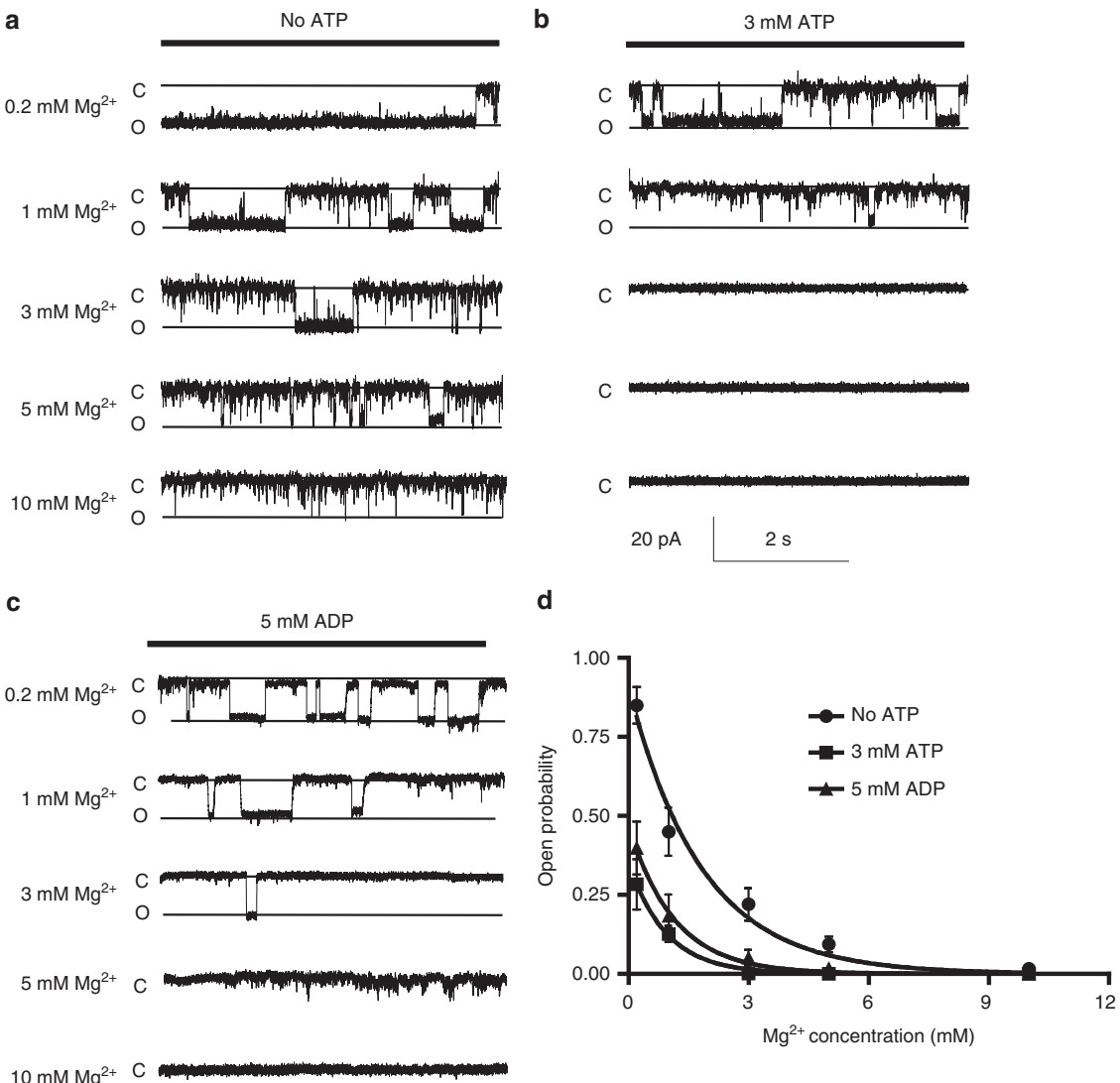

**Fig. 2** Patch clamp analyses of the MgtE-nucleotide complexes. **a–c** The MgCl$_2$ concentration in the bath was increased from 0.2 to 10 mM, and single currents were measured at each MgCl$_2$ concentration for wild type MgtE-reconstituted GUVs with **b** ATP, **c** ADP, and without **a** nucleotide. Representative current traces recorded at −120 mV from a single membrane patch at different [Mg$^{2+}$]$_{in}$ are shown. (**d**) The open probabilities at different [Mg$^{2+}$] with and without nucleotide were calculated (*Bars* represent±SEM, $n = 5$ for each condition). The Hill coefficients for apo, ATP, and ADP are 0.3, 1.0, and 0.4, respectively

previously reported full-length structure (Supplementary Fig. 3). Consistent with the previous structure, the electron density for Mg7 is relatively weak, possibly indicating the weak affinity of the Mg7 site for Mg$^{2+}$. We also determined the crystal structure of the cytosolic domain of MgtE in complex with ATP at 3.0 Å resolution (Fig. 2c, d, Supplementary Fig. 2c, d and Table 1). The overall structure of the cytosolic domain is essentially identical to that in the full-length MgtE structure in complex with ATP, with an RMSD value of 0.9 Å for 230 Cα atoms.

Since the cytosolic domain structure was solved at a higher resolution, we describe the ATP binding by MgtE mainly based on the structure of the cytosolic domain of MgtE in complex with ATP.

ATP molecules are located proximal to the subunit interface between the N domain in one subunit and the CBS domain in the neighboring subunit (Fig. 3a). While we observed the binding of Mg$^{2+}$ ions to the cytosolic domain, as reported in the previous MgtE structures[8, 16], the phosphate groups of ATP are 5.4 Å away from the closest Mg$^{2+}$ ion in the ATP-bound structure (Fig. 3b).

Therefore, ATP binding to MgtE may not be directly coupled with Mg$^{2+}$ binding.

The ATP molecule is exclusively recognized by the CBS domain (Fig. 3c, d). The adenine base of ATP is stabilized by three hydrogen bonds with the side chain of Asn203, and the main-chain carbonyl and amino groups of Val207, together with a π stacking interaction with Phe227 (Fig. 3d). To obtain structural insights into the base specificity of MgtE for ATP over GTP, we superimposed GTP onto ATP in the structure. In the superimposed model, the amino group of the guanine base clashes with the molecular surface of the ATP binding pocket, while there is no such steric hindrance between the adenine ring and MgtE (Supplementary Fig. 4).

The ribose moiety of ATP forms two hydrogen bonds with the side chain of Asp188 (Fig. 3d), while the α- and β-phosphates of ATP interact with the side chains of Tyr170 and Arg187, respectively (Fig. 3d). The phosphate groups of ATP adopt the extended conformation, and thus the three phosphate groups are exposed to the subunit interface between the N and CBS domains (Fig. 3c, d).

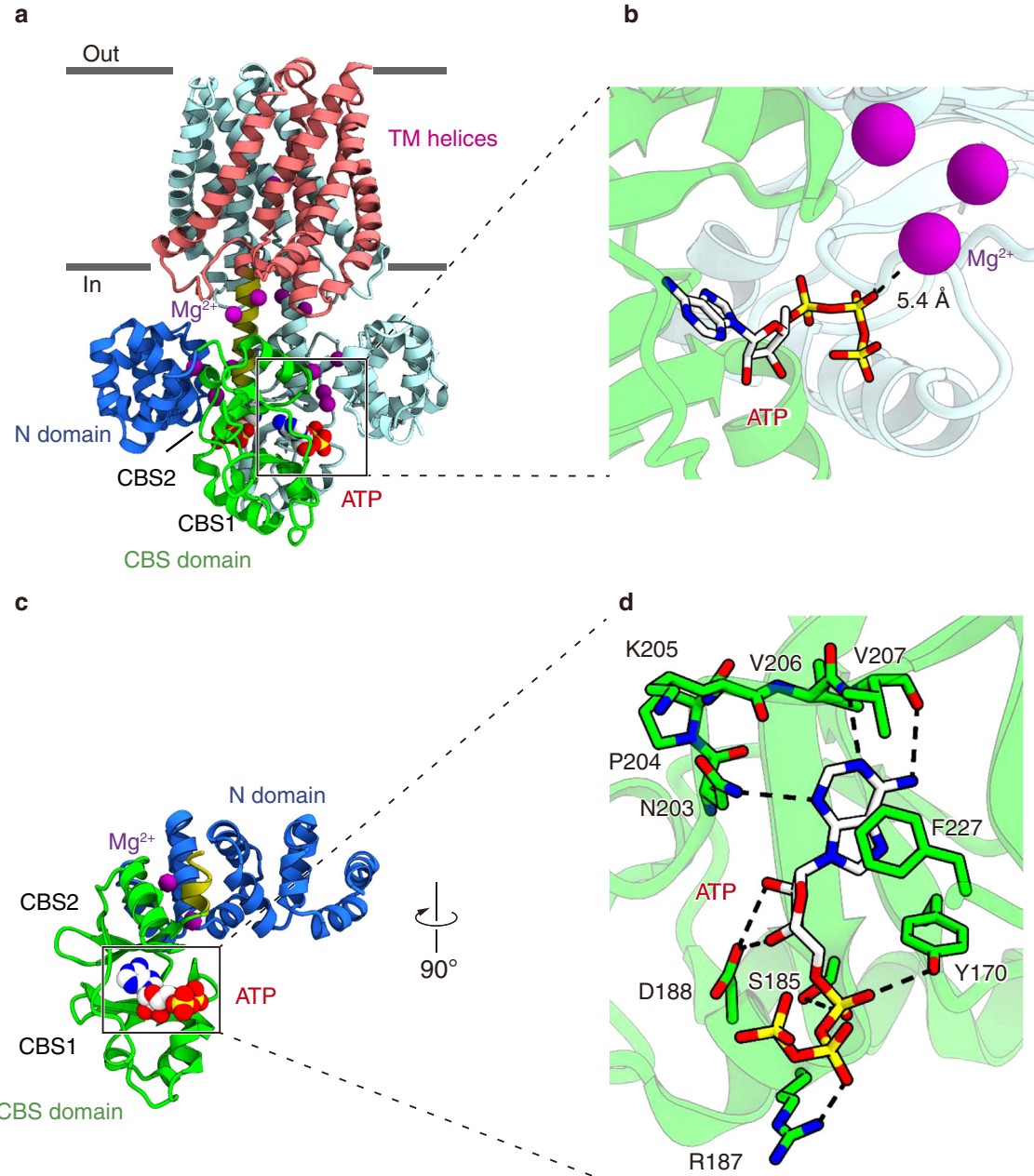

**Fig. 3** Overall structure and ATP binding site of MgtE. **a** Ribbon representation of the full-length MgtE structure in complex with ATP, viewed parallel to the membrane. The N domain, CBS domain, plug helix, and transmembrane helices in one subunit are colored *blue*, *green*, *yellow*, and *red*, respectively. ATP and Mg$^{2+}$ are shown as sphere models **a**, **c**. **b** Close-up view of the ATP binding site in the full-length structure. The *dotted black line* represents the minimum distance between the β-phosphate group of ATP and Mg$^{2+}$ (5.4 Å). ATP is shown as a stick model **b**, **d**. **c** Ribbon representation of the MgtE cytosolic domain structure in complex with ATP. **d** Close-up view of the ATP binding site in the MgtE cytosolic domain structure. Amino acid residues involved in ATP-binding are shown as stick models. *Dotted black lines* indicate the hydrogen bonds between MgtE and ATP

Notably, all of these residues involved in the ATP binding are highly conserved among the MgtE proteins, and this ATP binding motif is similarly conserved among the CBS domain-containing proteins, such as the human Cl- transporter ClC-5 (Supplementary Figs. 5 and 6).

**Mechanism of ATP-dependent Mg$^{2+}$ sensing by MgtE.** To elucidate the molecular mechanism of the modulation of the MgtE channel gating by ATP, we conducted biochemical, electrophysiological, and genetic analyses with structure-based

mutants of TtMgtE. We targeted the residues involved in the interactions with the phosphate groups (R187A, R187E), the ribose group (D188A), and the adenine ring (F227A).

We first performed the limited protease analysis of the MgtE mutants (Fig. 4a). While the results obtained with the D188A mutant were similar to those obtained with the wild type, the F227A mutant did not exhibit the band corresponding to the full-length protein under the 32 mM Mg$^{2+}$ and 3 mM ATP conditions, unlike the wild type (Fig. 4a). This result suggested that the F227A mutant is unable to bind to ATP. Intriguingly, the R187A and R187E mutants exhibited the band corresponding to

**Table 1 Data collection and refinement statistics**

|  | Full-length | Cytosolic domain |
| --- | --- | --- |
| *Data collection* | | |
| Wavelength (Å) | 0.9791 | 0.9791 |
| Space group | $C\,2$ | $P2_12_12_1$ |
| Cell dimensions | | |
| $a$, $b$, $c$ (Å) | 131.6, 83.3, 152.3 | 138.3, 106.7, 90.2 |
| $\alpha$, $\beta$, $\gamma$ (°) | 90, 100.0, 90 | 90, 90, 90 |
| Resolution (Å)[a] | 50.3-3.61(3.82-3.60) | 48.80-2.99 (3.17-2.99) |
| $R_{sym}$[a] | 0.072 (1.432) | 0.109 (0.567) |
| $I/\sigma I$[a] | 15.09 (1.70) | 10.73 (2.71) |
| Completeness (%)[a] | 98.5 (98.6) | 97.5 (95.4) |
| Redundancy[a] | 6.8 (6.8) | 6.5 (6.6) |
| $CC_{1/2}$ (%)[a] | 99.9 (79.7) | 99.8 (94.6) |
| *Refinement* | | |
| Resolution (Å) | 3.6 | 3.0 |
| No. reflections | 20,361 | 26,821 |
| $R_{work}$/$_{free}$ | 25.0/28.3 | 24.0/29.8 |
| No. atoms | | |
| Protein | 6522 | 7956 |
| ATP | 62 | 124 |
| $Mg^{2+}$ | 13 | 8 |
| B-factors | | |
| Protein | 169.8 | 73.3 |
| ATP | 152.8 | 81.2 |
| $Mg^{2+}$ | 127.7 | 48.3 |
| R.m.s deviations | | |
| Bond lengths (Å) | 0.010 | 0.011 |
| Bond angles (°) | 1.519 | 1.559 |
| Ramachandran plot | | |
| Favoured (%) | 93.99 | 98.27 |
| Allowed (%) | 6.01 | 1.73 |
| Outliers (%) | 0 | 0 |

[a]Highest resolution shell is shown in parentheses

the full-length protein under the 32 mM $Mg^{2+}$ conditions even in the absence of ATP, whereas the wild type protein did not in the absence of ATP (Fig. 4a). In addition, the intensity of the band corresponding to the full-length protein in the R187A mutant is weaker than that in the R187E mutant (Fig. 4a). This result implied that the loss of the positive charge and the introduction of the negative charge in the side chain of R187 may have a similar effect to that of the binding of ATP with negatively charged phosphate groups.

To further investigate the roles of the adenine ring (F227A) and the phosphate group (R187E), we performed biochemical, electrophysiological, and genetic analyses of the F227A mutant and the R187E mutant. The injection of ATP into the F227A mutant solution showed no heat of binding by ITC (Fig. 4b), demonstrating that the F227A mutant lacks the ability to bind to ATP, consistent with the limited protease analysis (Fig. 4a). The injection of ATP into the R187E mutant solution still showed the weaker binding to ATP, as compared to the binding of the wild type to ATP (Fig. 4b). Therefore, this result suggested that the side chain of Arg187 is involved in the recognition of the γ-phosphate group of ATP, but is not requisite for the ATP binding ability. Consistently, ADP, which lacks the γ-phosphate group, bound to MgtE, but with lower affinity than that of ATP (Fig. 1a).

Next, we performed the patch clamp analysis of the F227A and R187E mutants, using proteoliposomes under 0–10 mM $[Mg^{2+}]_{in}$ conditions, with 0 or 3 mM ATP (Fig. 5).

Unlike the wild type, we did not observe the ATP effect with the F227A mutant. Altogether with the ITC analysis, these results demonstrated that ATP binding to the MgtE cytosolic domain, rather than ATP binding to $Mg^{2+}$, indeed induced the

modulation of the channel gating observed in the wild type MgtE (Fig. 5a, c). Furthermore, the Arg187 mutant generated similar plots of the open probabilities in the presence and absence of ATP, resembling that of the wild type MgtE under the 3 mM ATP conditions (Fig. 5b, d). These results indicated that the loss of the positive charge and the introduction of the negative charge at the position of Arg187 had a similar effect to that of the binding of negatively charged ATP.

Considering the proposal that the positively charged Arg187 recognizes the negatively charged phosphate group of ATP, this result suggested that the negative charges derived from the phosphate groups of ATP might allow the MgtE cytosolic domain to attract more positive charges, thus tuning the affinity of the MgtE cytosolic domain for $Mg^{2+}$ within a physiological range. To further test this hypothesis, we evaluated the effect of ADP, lacking the γ-phosphate group, by the patch clamp analysis of MgtE under conditions with 0–10 mM $[Mg^{2+}]_{in}$ and 5 mM ADP (Fig. 2). In the presence of 5 mM ADP in the bath solution, the channel was still active with 3 mM $[Mg^{2+}]_{in}$, while the channel was completely inactivated with 5 mM $[Mg^{2+}]_{in}$ (Fig. 2c). Therefore, ADP also enhances the affinity of MgtE for $Mg^{2+}$, but has a smaller effect than that of ATP (Fig. 2), which is consistent with our hypothesis.

Furthermore, to test whether MgtE possesses ATPase activity, we performed an ATP hydrolysis assay (Supplementary Fig. 7). In the ATP hydrolysis assay, we did not detect ATPase activity with the wild type MgtE, as compared with apyrase, an ATP-diphosphohydrolase. In addition, the level of ATP hydrolysis activity of the wild type MgtE is the same as that of the F227A mutant, which lacks ATP binding activity (Supplementary Fig. 7). Accordingly, while ATP can bind to MgtE to modulate the channel gating, MgtE either lacks or has very low ATPase activity. This conclusion is also supported by the crystal structures of the MgtE–ATP complex, where the γ-phosphate group of ATP is clearly observed (Supplementary Fig. 2).

**ATP-binding site contributes to cellular $Mg^{2+}$ homeostasis.** To characterize the significance of the ATP binding site in vivo, we conducted genetic analyses of ATP-binding site mutants (Fig. 6). Due to their similar coordination chemistry, $Mg^{2+}$ channels/transporters also typically permeate $Co^{2+}$ and $Ni^{2+}$ [6]. Thus, the change in the sensitivities to $Co^{2+}$ and $Ni^{2+}$ has been employed to characterize the $Mg^{2+}$ transport system[6]. The increased sensitivities to $Co^{2+}$ and $Ni^{2+}$ indicate excess $Co^{2+}$ and $Ni^{2+}$ uptake, as they are toxic. The expression of MgtE in *E. coli* also enhances the susceptibility to $Co^{2+}$ and $Ni^{2+}$, by transporting these ions into cells[8]. In particular, the expression of the cytosolic domain $Mg^{2+}$-binding site mutants in *E. coli* caused a strong dominant-negative effect on the growth in $Co^{2+}$-containing media and $Ni^{2+}$-containing media. There results suggested that the mutants exhibited hypersensitivities to $Co^{2+}$ and $Ni^{2+}$, and indirectly demonstrated the loss of the intracellular $Mg^{2+}$-dependent control of $Mg^{2+}$ uptake by MgtE[8]. Similarly, we conducted the $Co^{2+}$ and $Ni^{2+}$ sensitivity assays with the ATP binding site mutants of MgtE, to indirectly evaluate the effects of these mutations on the regulatory function of MgtE for $Mg^{2+}$ homeostasis (Fig. 6). Intriguingly, the R187E and F227A mutants also showed $Co^{2+}$ and $Ni^{2+}$ hypersensitivities similar to that of the $Mg^{2+}$-binding site mutant (ΔN) (Fig. 6b), whereas the expression of these ATP-binding site mutants complemented the $Mg^{2+}$ auxotrophic *E. coli* strain lacking the major set of genes encoding $Mg^{2+}$ transporters (Fig. 6a). Therefore, these results indicated that the ATP-dependent modulation of MgtE is also important for MgtE to maintain the cellular $Mg^{2+}$ homeostasis in vivo.

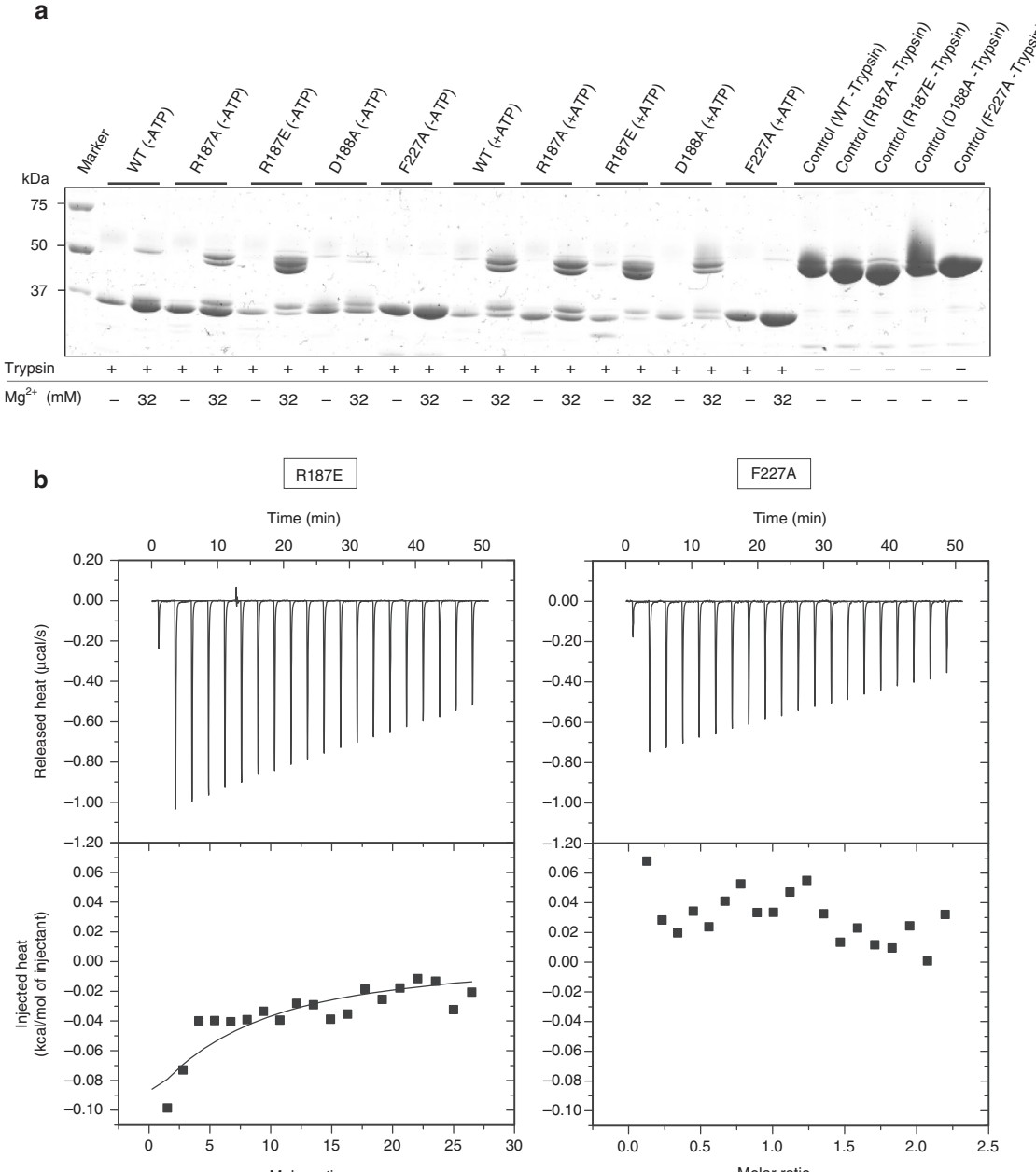

**Fig. 4** Biochemical analyses of the ATP-binding site mutants of MgtE. **a** Protease protection of the MgtE mutants by $Mg^{2+}$ and ATP. Protease protection of wild type MgtE by $Mg^{2+}$ and ATP is shown as a reference. The numbers on the left side indicate the molecular masses (in kilodaltons) of the markers. **b** ITC data of the F227A and R187E mutants with ATP. The raw ITC data and the plots of injected heat for 20 automatic injections of 10 mM nucleotide solution into the sample cell containing the MgtE solution are shown. Measurements were repeated twice, and similar results were obtained

## Discussion

In this work, we demonstrated that ATP binds to MgtE and modulates its $Mg^{2+}$-dependent channel gating (Figs. 1 and 2). The crystal structure of the MgtE–ATP complex and the structure-based mutational analysis revealed further structural and functional insights into the modulation mechanism of the MgtE channel gating by ATP (Figs. 3–7).

In the presence of ATP, the MgtE cytosolic domain exhibited higher affinity for $Mg^{2+}$ within a physiological range (Figs. 2b, d, and 7b), which would enable the MgtE cytosolic domain to act as a $Mg^{2+}$ sensor in vivo. These results provide an explanation for the huge discrepancy between the physiological intracellular $Mg^{2+}$ concentration and the previous electrophysiological analysis of the $Mg^{2+}$-dependent gating of MgtE[8].

In contrast, in the absence of ATP, the threshold of intracellular $Mg^{2+}$ for the channel inactivation was much higher than the physiological $Mg^{2+}$ concentration (Fig. 2). Instead, the dissociation of ATP from MgtE facilitated the $Mg^{2+}$ influx into cells, even at high concentrations of intracellular $Mg^{2+}$ (Figs. 2 and 7a). In other words, ATP switches the gating mode of MgtE from the $Mg^{2+}$ sensor mode (with ATP) to the upregulated mode (without ATP). Intriguingly, the cytosolic ATP level seemed to correlate with the cytosolic $Mg^{2+}$ level in bacteria[23] and mammalian cells[24], and it is well known that $Mg^{2+}$ ion plays a vital role in ATP synthesis[25]. In addition to ATP synthesis, $Mg^{2+}$ and ATP are functionally coupled in many physiological processes[26,27]. Therefore, it is physiologically reasonable that MgtE exhibits higher channel activity in the absence of ATP.

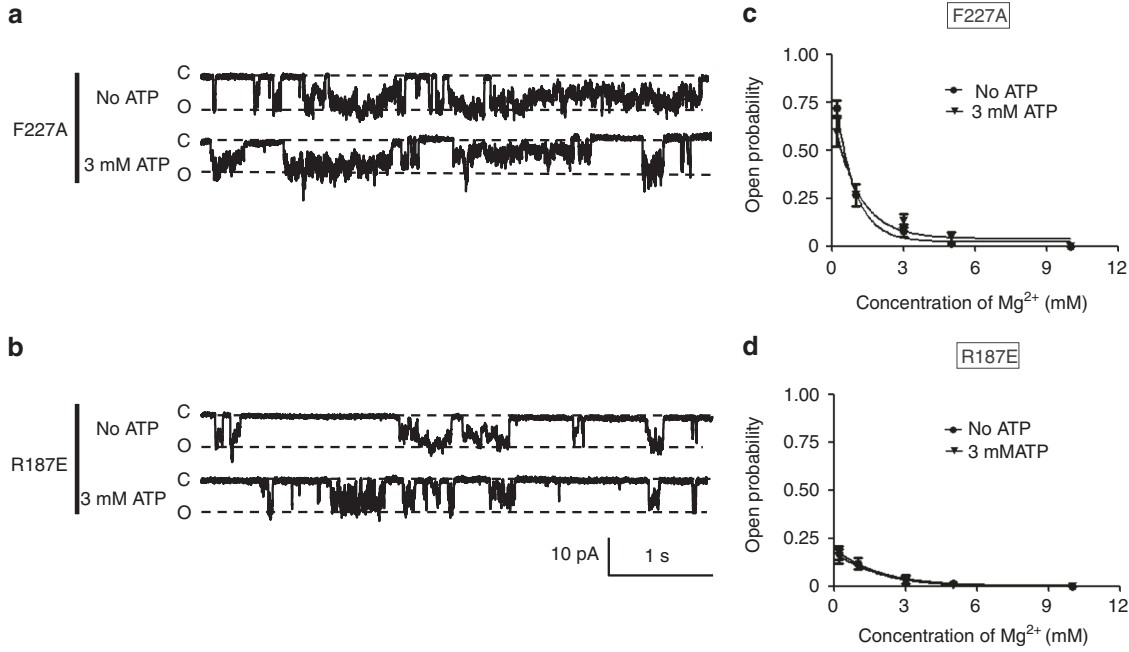

**Fig. 5** Patch clamp analyses of ATP-binding site mutants of MgtE. **a**, **b** Representative current traces recorded at −120 mV from a single membrane patch at 0.2 mM of $Mg^{2+}$, with or without ATP, by the MgtE mutants F227A **a** and R187E **b**. **c**, **d** The open probabilities were determined at different $[Mg^{2+}]_{in}$ with or without ATP with the MgtE mutants F227A **c** and R187E **d** (*Bars* represent ±SEM, n = 5 for each condition)

In addition to MgtE, multiple channels/transporters containing a CBS domain have been identified[28, 29]. In particular, the ClC Cl⁻ channels/transporters have been functionally and structurally well characterized, and nucleotide binding to the CBS domains of some ClC channels/transporters, such as ClC-1 and ClC-5, modulates their transport activities[30]. In the full-length structures of the eukaryotic ClC channel/transporter, the latter part of the tandemly repeated CBS domains, CBS2, faces toward the transmembrane domain[31, 32]. It is hypothesized that such contacts might allow the CBS domain to induce the structural changes of the transmembrane domain[30]. Similarly, the interactions between the CBS and transmembrane domains of MgtE are mediated largely by the CBS2 region (Fig. 3), which undergoes a large structural change presumably upon the channel gating[8, 16, 18]. Furthermore, as we observed in MgtE (Fig. 2), the number of phosphate groups in the nucleotides also plays an important role in the regulation of *Arabidopsis thaliana* ClC[33]. It is intriguing that the MgtE and ClC proteins, two distinct CBS domain-containing channels/transporters, possess these common features.

Our work also revealed the novel feature of MgtE, among the CBS domain-containing channels/transporters. A characteristic of MgtE is that its channel activity is modulated by both $Mg^{2+}$ and ATP. Like MgtE, the CorA family of $Mg^{2+}$ channels is also inactivated upon intersubunit $Mg^{2+}$ binding to the cytosolic domain[34–36], but its members possess neither a CBS domain nor any other known ATP binding motif. Recently, another class of $Mg^{2+}$ channels/transporters, CBS domain divalent metal cation transport mediators (CNNMs), was reported to have $Mg^{2+}$-dependent ATP binding activity mediated by the CBS domain[37]. CNNM transporters are thought to function as either influxers or effluxers, and $Mg^{2+}$ or $Mg^{2+}$-ATP might function as a modulator of CNNM transporters[38–40]. Therefore, although it remains unclear whether or how the $Mg^{2+}$-dependent ATP binding modulates the $Mg^{2+}$ permeation by CNNM proteins, it is attractive to speculate that the CNNM $Mg^{2+}$ transporters are also modulated by both ATP and $Mg^{2+}$

for cellular $Mg^{2+}$ homeostasis, as we showed in this work on MgtE.

In conclusion, our analyses have revealed the molecular basis for the ATP-dependent modulation of MgtE for $Mg^{2+}$ homeostasis, and thus provided the missing link between the cytosolic ATP and $Mg^{2+}$ levels in the regulation of $Mg^{2+}$ channels.

## Methods

**Purification and crystallization**. The full-length *Thermus thermophilus* MgtE (TtMgtE) was overexpressed in *Escherichia coli* C41 (DE3) cells and solubilized with n-dodecyl-β-maltoside (DDM), as described previously[16]. After ultra-centrifugation, the solubilized supernatant was applied to a Ni-NTA (Qiagen) column preequilibrated with buffer A (50 mM HEPES, pH 7.0, 150 mM NaCl), containing 0.1% (w/v) DDM and 20 mM imidazole. The column-bound proteins were washed with buffer A containing 0.25% (w/v) n-nonyl-β-D-thiomaltoside (NTM) and 50 mM imidazole, and eluted with buffer A containing 0.25% (w/v) NTM and 300 mM imidazole. The eluted MgtE fractions were pooled, concentrated with an Amicon Ultra 50 K filter (Millipore), and then applied to a Superdex 200 10/300 size-exclusion column (GE Healthcare), equilibrated with buffer B (20 mM HEPES, pH 7.0, 150 mM NaCl, 0.25% (w/v) NTM). The purified protein was concentrated to ~ 12 mg/ml, using an Amicon Ultra 50K filter. After concentration, the protein was mixed with 1/10 volume of buffer B containing 100 mM ATP, and incubated at 4 °C for 1 h. The crystal was obtained by vapor diffusion over a solution containing 9% PEG 4000, 0.2 M MgCl₂, and 0.05 M MES, pH 6.5. Before cryocooling, the crystals were transferred into a cryoprotectant solution containing 10 mM ATP, 9% PEG 4000, 0.2 M MgCl₂, 0.05 M MES, pH 6.5, and 30% (w/v) PEG 400. The cytosolic domain (1–275) of MgtE was overexpressed in *Escherichia coli* C41 (DE3) and purified, as described previously[41]. The purified protein was concentrated to ~ 15 mg/ml, and was mixed with 1/10 volume of the gel filtration buffer containing 100 mM ATP at 4 °C for 1 h. The crystals were obtained by vapor diffusion over a solution containing 20–22% PEG 400, 0.2 M MgCl₂, and 0.1 M HEPES, pH 7.4. Before cryocooling, the crystals were transferred into a cryoprotectant solution containing 10 mM ATP, 33% PEG 400, 0.2 M MgCl₂, and 0.1 M HEPES, pH 7.4.

**Data collection and structure determination**. All data sets were collected at the SPring-8 BL41XU (Hyogo, Japan). The data sets were processed with the XDS programs[42]. The phases of the full-length MgtE–ATP complex and the cytosolic domain of the MgtE–ATP complex were obtained by molecular replacement with the CCP4 suite programs[43], using the previously determined structures of the full-length MgtE (PDB: 2ZY9) and the cytosolic domain of MgtE (PDB: 2YVV) as

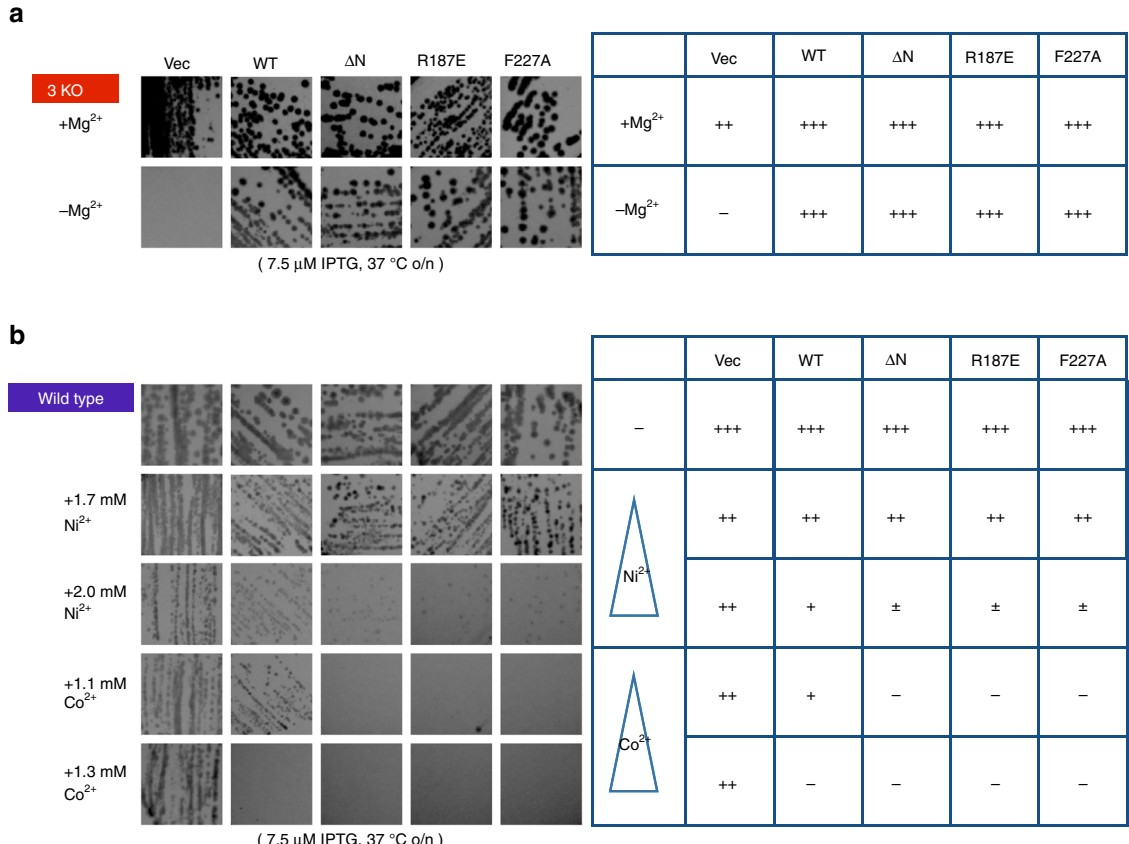

**Fig. 6** *E. coli* growth assay of ATP-binding site mutants. **a** $Mg^{2+}$-auxotrophic growth complementation assay. The $Mg^{2+}$-auxotrophic *E. coli* strain, transformed with each plasmid, was streaked on LB or LB with 100 mM $MgSO_4$. **b** $Co^{2+}$ and $Ni^{2+}$ sensitivity assays. The wild type *E. coli* strain (W3110 DE3), transformed with each plasmid, was streaked on LB or LB with 1.7 mM $Ni^{2+}$, 2.0 mM $Ni^{2+}$, and either 1.1 mM $Co^{2+}$ or 1.3 mM $Co^{2+}$, and the colony growth was monitored. The growth of each transformant is indicated as "+++" (similar to that of empty vector), "++" (less), "+" (scarcely any), "±" (severe growth retardation) and "−" (no growth at all)

the search models, respectively. The models were subsequently improved through iterative cycles of manual building with COOT[44] and refinement with the program PHENIX[45]. The structure refinement statistics are summarized in Table 1. Molecular graphics were illustrated with CueMol (http://www.cuemol.org).

**Limited Proteolysis**. The full-length WT and mutant MgtE proteins were prepared as described above. The 9 µl reaction-premix contained 4 µl of the full-length MgtE (2 mg/ml), 1 µl of 0–32 mM $MgCl_2$, and either 1 µl gel filtration buffer and 3 µl 10 mM ATP pH 7.0 (ATP+) or 4 µl of gel filtration buffer. After an incubation at 4 °C for 1 h, 1 µl of 16 µg/ml trypsin was added to each reaction-premix, and the reactions proceeded at 4 °C for 17 h. After the addition of 10 µl of SDS-PAGE sample buffer, the samples were boiled and immediately subjected to SDS-PAGE on 12.5% gels.

**Isothermal titration calorimetry analysis**. The binding of ATP and other nucleotides to MgtE and its mutants was measured using a MicroCal ITC 2000 microcalorimeter (GE Healthcare) at 20 °C. The full-length MgtE and its mutants were purified as described above. The peak fractions from the gel filtration were collected and diluted to 0.06–0.09 mM as a monomer (3–4 mg/ml) with the gel filtration buffer. The ligand solutions used for titration were prepared by adding ATP or the ATP analog to the gel filtration buffer, at a final concentration of 10 mM. The pH of the ligand solutions was adjusted to 6.5, by adding 4 M NaOH. For the measurement in the presence of $Mg^{2+}$, purified MgtE was dialyzed and diluted with gel filtration buffer containing 10 mM $Mg^{2+}$. The ligand solution was prepared by adding ATP to the gel filtration buffer containing a final concentration of 10 mM $Mg^{2+}$. The pH of the ligand solution was adjusted to 6.5, by adding 4 M NaOH. The ligands were injected 20 times (0.4 µl for injection 1, 2 µl for injections 2–20), with 150 s intervals between injections. The background data obtained from the buffer sample were subtracted before the data analysis. The data were analyzed with the Origin7 software package (MicroCal). Measurements were repeated twice, and similar results were obtained.

**Patch clamp analysis**. The giant unilamellar vesicles (GUVs) reconstituted with the target protein (protein:lipid = 1:1000, wt:wt; lipid, azolectin (Sigma)) were prepared by a modified sucrose method[46]. First, 200 µl of a 25 mg/ml solution of azolectin in chloroform was dried in a glass test tube under a stream of $N_2$ while rotating the tube, to produce a homogeneous dried lipid film. Subsequently, 1 ml of 0.4 M sucrose was placed at the bottom of the tube, and the solution was incubated at 50 °C for 1–2 h until the lipid was resuspended. After cooling the solution to room temperature, the purified proteins were added to achieve the desired protein-to-lipid ratio. The glass tube containing the protein–lipid solution was shaken gently on an orbital mixer for three hours at 4 °C. After this procedure, the sample was ready for patch clamping. The pipette buffer contained 210 mM N-methyl-D-glucamine, 90 mM $MgCl_2$, and 5 mM HEPES (pH 7.2). The bath buffer contained 300 mM N-methyl-D-glucamine and 5 mM HEPES (pH 7.2). The data were acquired at −120 mV at a sampling rate of 20 kHz with a 5-kHz filter, using an AxoPatch 700B amplifier in conjunction with the pClamp 10 software (Axon Instruments). $Mg^{2+}$ binding to the MgtE cytosolic domain reportedly inhibited the channel opening completely, at concentrations over 10 mM. Accordingly, in this experiment, only the MgtE protein oriented with the cytosolic domain facing toward the bath solution side contributes to the current recording of MgtE, and the MgtE with the opposite orientation does not contribute to the current recording, since all tested pipette solutions included 90 mM $MgCl_2$. Multiple channel openings were occasionally observed, and ATP had essentially same effect on the MgtE activity in such recordings as well.

**Quantitative analysis of the bands in the SDS-PAGE gel**. The intensities of the bands in the SDS-PAGE gel were quantified using the ImageJ software[47]. Each band in the SDS-PAGE gel was selected manually, and the intensity peak was obtained and integrated using the "magic wand" tool in ImageJ. In the limited proteolysis experiment (Fig. 4), the relative intensity of the full-length band of the R187A mutant was 0.57, as compared to that of the R187E mutant.

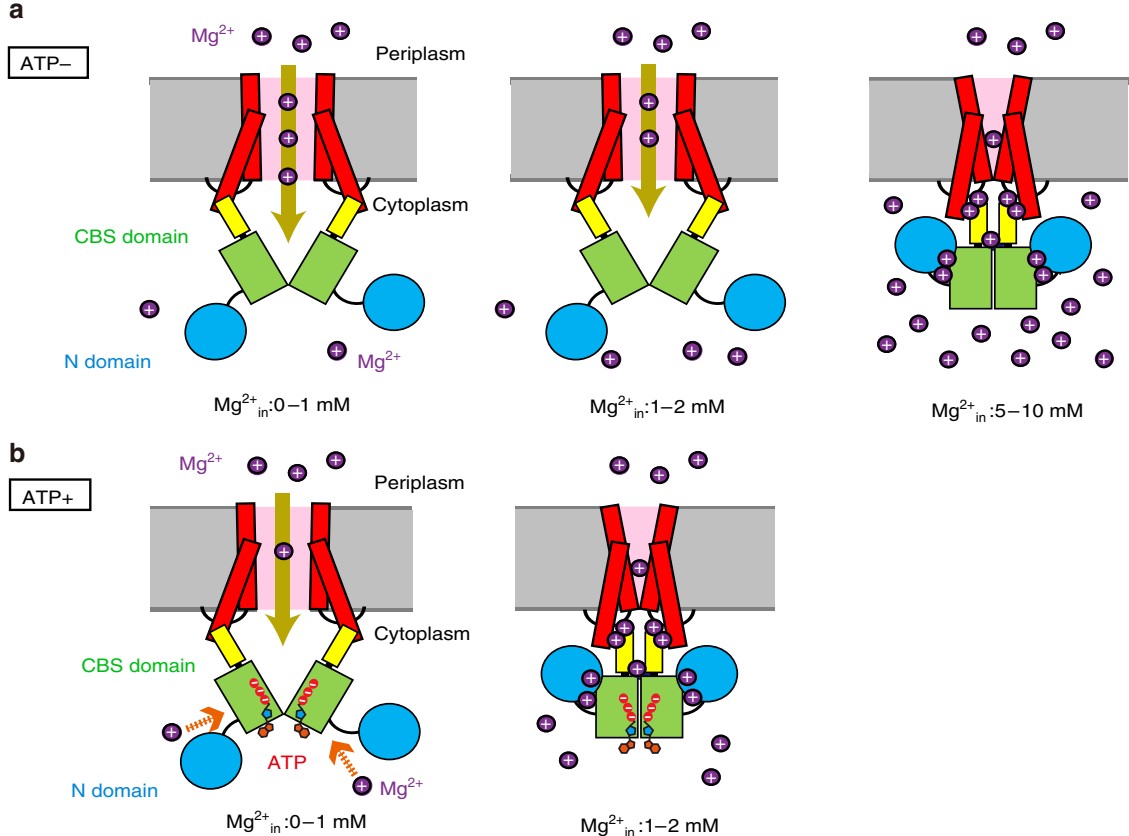

**Fig. 7** Proposed functional role of intracellular ATP in the $Mg^{2+}$-dependent gating of MgtE. A cartoon model of the proposed mechanism. The coloring scheme is the same as in Fig. 2. The *brown arrows* indicate the $Mg^{2+}$ influx into the cells. Without ATP binding to the cytosolic domain, MgtE still opens when the intracellular $Mg^{2+}$ concentration ($Mg^{2+}_{in}$) is above the physiological range (1–2 mM), and closes only at very high [$Mg^{2+}_{in}$] (5–10 mM) **a**. With ATP bound to the cytosolic domain, MgtE closes when the intracellular $Mg^{2+}$ concentration is above the physiological range (1–2 mM) **b**. The *orange arrows* indicate the ATP-dependent attraction of $Mg^{2+}$ to the cytosolic domain

**E. coli in vivo assays.** In the magnesium requirement growth complementation assays, the $Mg^{2+}$-auxotrophic strain (BW25113 $\Delta mgtA$ $\Delta corA$ $\Delta yhiD$ DE3[8]) was transformed with the plasmids, and transformants were obtained on LB (+50 µg/ml kanamycin) plates supplemented with 100 mM $MgSO_4$. Each of the transformant colonies was streaked on LB±$Mg^{2+}$ plates and incubated at 37 °C overnight. In the Ni/Co sensitivity assays, the wild type strain (W3110 DE3[8]) was transformed with the plasmid, and transformants were obtained on LB (+50 µg/ml kanamycin) plates. Each of the transformant colonies was then streaked on an LB (+50 µg/ml kanamycin) plate, as well as plates supplemented with $NiCl_2$ and $CoCl_2$ at the indicated concentrations, and incubated at 37 °C overnight. Both assay experiments were performed in the presence of 20 µM IPTG, for optimal MgtE expression.

**ATP hydrolysis assay.** ATP hydrolysis assays were performed using the malachite green method[48]. The malachite green dye solution, containing 36 ml of 0.045% malachite green, 12 ml of 4.2% ammonium molybdate, and 1 ml of 1% Triton X-100, was freshly prepared on the day of the experiment. The assays were performed in a final reaction mixture consisting of the basal buffer (20 mM HEPES-NaOH (pH 7.0), 20 mM $MgCl_2$, 150 mM NaCl, 0.05% DDM) and 2 mM ATP, with and without 1.25 µM MgtE or apyrase proteins. To initiate the reaction, 7.5 µl of ATP was added to 367.5 µl of the pre-reaction mixture. The reaction was performed at RT for 1 h. Afterwards, 800 µl of the malachite green dye solution, 50 µl of the basal buffer, 50 µl of each reaction mixture and 100 µl of 34% citric acid were mixed, and then the absorbance was measured at 660 nm. The absorbance standard curve for inorganic phosphate was established with standard $H_3PO_4$ solutions.

**Data availability.** The atomic coordinates and structure factors for the full-length MgtE in complex with ATP and the MgtE cytosolic domain in complex with ATP have been deposited in the Protein Data Bank, under the accession codes 5X9H and 5X9G, respectively. All other data are available from the corresponding authors upon reasonable request.

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

## Acknowledgements

We thank the Nureki lab members, especially Tomohiro Nishizawa and Go Kasuya (University of Tokyo), for critical comments on the manuscript, the beam-line staff at BL41XU of SPring-8 (Harima, Japan) for technical assistance during data collection, and Prof. Ye Yu (Shanghai Jiao Tong University) for the initial patch clamp recording trials of MgtE with HEK293 cells. The diffraction experiments were performed at SPring-8 BL41XU. This work was supported by funds from the Ministry of Science and Technology of China (2016YFA0502800) to M.H. and Z.Y., and the National Natural Science Foundation of China (projects 31571083, 31570838, and 31650110469), the Platform for Drug Discovery, Informatics and Structural Life Science of the Ministry of Education, Culture, Sports, Science and Technology (MEXT), JSPS KAKENHI (Grant Nos. 24227004, 16H06294, 25291011), the FIRST program, PRESTO, a Grant-in-Aid for JSPS Fellows, and a grant from the Young 1000 Talent Program (2015) to Z.Y. and M.H. Additional support was provided to Z.Y. by The Program for Professor of Special Appointment (Eastern Scholar of Shanghai, TP2014008) and The Shanghai Rising-Star Program (4QA1400800).

## Author contributions

A.T. expressed and purified MgtE and its mutants for crystallization, and determined the structures with assistance from H.T., R.I., and M.H. A.T. performed the limited proteolysis and the ITC analysis, with assistance from T.M., M.O., and I.S. M.Z. and W.Z. performed the patch-clamp analysis of MgtE. F.J. expressed and purified MgtE and its mutants for the patch-clamp analysis and the ATPase activity assay, and conducted the ATPase activity assay of MgtE. N.D. conducted the mass-spectrometry. K.I. performed the in vivo complementation assay. K.H. and H.K. contributed to the early patch clamp analysis trials of MgtE with giant *E. coli* spheroplasts. A.T., M.H., Z.Y., and O.N. wrote the manuscript. Z.Y., M.H., and O.N. supervised the research.

## Additional information

**Competing interests:** The authors declare no competing financial interests.

