## [Peer Review File · Nature Communications]

Reviewers' Comments:

Reviewer #1 (Remarks to the Author)

In their manuscript, the authors describe an interdisciplinary study that characterizes the regulation of the Mg²⁺-selective channel MgtE by ATP. To address this question, they provide data from different experimental techniques including X-ray crystallography, titration calorimetry, limited proteolysis, electrophysiology and *in vivo* studies. The data appears of high quality and the results all support the hypothesis that the binding of ATP shifts the previously described inhibition of the channel by intracellular Mg²⁺ towards physiological concentrations. The same group has previously determined the first structure of the MgtE channel and has characterized its functional properties. This manuscript provides another very important contribution towards the understanding of the structural and functional properties of this important protein family. I thus think that it is an excellent candidate for publication in *Nature Communications* after some revisions.

I have several remarks the authors should consider:

Crystallography:

- I did not have access to the table containing data collection and refinement statistics since it was not included in the files for review. The reported resolution of the full-length channel is 3.6 Å and it is not clear whether this estimate is based on a classical $I/\sigma I$ criterion or by CC1/2 that usually increases the nominal high-resolution limit by about 0.2 Å. In any case, the resolution is on the lower side for a detailed atomistic interpretation of the density. This likely creates no problem in this case since the structure at high resolution in complex with Mg²⁺ is known and the described ATP/Mg²⁺ complex adopts a similar conformation. The large ATP ligand can also be identified at this resolution and is supported by the ATP binding observed in the higher resolution structure of the cytoplasmic domains. Still, I think that the authors should carefully document the quality of their structures in the supplemental materials by showing stereo figures of the refined 2Fo-Fc density of the entire protein and close-ups of relevant regions. I also think that the Table containing the data collection and refinement statistics should be part of the manuscript.
- The comparably low resolution of the data makes a definite assignment of Mg²⁺ positions ambiguous. Do the authors think that the site for Mg²⁺ 7 shown in Supplementary Fig. 3 is occupied by the ion at all?

Electrophysiology:

- The authors show single channel traces from patches of giant unilamellar liposomes. I believe these represent whole liposome currents and not currents from excised patches. I wonder whether at the describe protein to lipid ratio the authors expect at most one channel per liposomes or whether they also have observed patches containing several channels. In general, it would be interesting to know which fraction of patched liposomes did contain a MgtE channel.
- Did the authors ever reconstitute at higher protein to lipid ratio and did they ever observe macroscopic currents with similar behavior.
- The authors describe that they have used 5 independent recordings to determine the open probabilities shown in Fig. 2c and 5c,d. What was the duration of each recording and were they from different reconstitutions?
- Which equation was used to fit the curves in Fig. 2c, 5c and d. Is there any indication for cooperativity in the inhibition by Mg²⁺ as evidenced by a Hill slope above 1?

Biochemistry:

- Was Mg²⁺ present in calorimetry experiments, and is there any Mg²⁺ dependence of ATP or ADP binding?
- The mutation of R187 removes a potential Trypsin cleavage site. Did this affect the results of the limited proteolysis experiments?

General:

- The legends should contain more details of the described figures.
- The secondary structure elements in close-up figures are barely visible.
- I do not quite understand the meaning of the brown arrows in Figure 7. I expect that the total influx of Mg²⁺ is dependent on its electrochemical potential and the number and open probabilities of the MgtE channels, given that there is only a single open state with one distinct conductance. If this is the case, why do arrows have a different thickness?

Reviewer #2 (Remarks to the Author)

In this manuscript, Nureki and coworkers present compelling experiments that ATP and Mg are synergistic effectors of channel opening of MgtE. It has long been established that Mg binding the cytosolic domain is a negative regulator of channel opening, stabilizing the closed conformation of the pump. However, the concentration of Mg required to see these effects in vitro appeared to be too high and not physiological. The authors establish that in the presence of what appears to be physiological ATP concentration, the concentration of Mg required to close the channel is 1-2 mM, or near physiological. The work is novel in that it provides a reasonable clear view of the mechanism of gating by an important metal transporter.

The authors show that ATP binds to the CBS domain, and structurally characterize this complex in the presence of bound Mg (up to 7 Mg bind to intact transporter, with at least five in the cytoplasmic domain). Interestingly, in their highest resolution structures, the Mg ion that makes closest approach to ATP is some 5.5 Å away; thus, the authors conclude that ATP and Mg binding are “cooperative” in some way.

Mutant transporters were constructed in an effort to validate key features of the ATP binding site in the CBS domain, with most of the work focused on the F227A and R187R mutants. F227 forms a pi-stacking interaction with the purine ring, with R187 makes close approach to the beta- and gamma-phosphate groups of ATP. F227A mutant fails to bind ATP and is nonfunctional; the R187E mutant however, appears to mimic the effects of ATP, via a poorly defined electrostatic model.

The work is thorough and the conclusions largely supported by the data; the finding of an accessory role of ATP binding in channel function is not new (see ref. 20), but is interesting nonetheless.

The authors may wish to consider the following comments:

1) The way in which the R187E mutant works requires additional study. The authors seem to imply that the gamma-phosphate of ATP provides an electrostatic environment that screens positive charge thus “attracting” Mg to the cytoplasmic domain to regulate opening. In this way the R187E mutant “mimics” the effect of ATP binding. To confirm this, the authors need to measure Mg binding curves in the presence vs. absence of ATP. The expectation is the Mg will bind more tightly to the mutant than to wild-type MgtE. Also, the authors assume that the rate of ATP hydrolysis is slow- was this checked?

- 2) The R187A mutant had the same characteristics of the R187E mutant in a limited proteolysis experiment (Fig. 4). This is incompatible with the speculation in point (1).
- 3) The authors need to cite other literature that measures ATP binding affinities to a CBS domain to establish that the binding affinities here are in fact physiological. I would also like to see Mg binding experiments as well, by ITC (see point 1); alternatively the authors could cite the relevant literature.
- 4) In order to enhance the impact of the work, the authors should revise the Discussion (which is very short and basically recapitulates the Results) and place their findings with MgtE in the context of known mechanisms of metal- or nucleotide-dependent gating in other metal transporter systems. Are there general trends?
- 5) The data in Fig. 6 need to be better described. It is unclear why the authors used Co and Ni poisoning as a proxy for Mg transporter function. Why not measure Mg directly in these cells. A better explanation of this experiment may be all that is required.
- 6) The manuscript requires a thorough editing. For example, p. 13, line 7, "heat of linkage" should be "heat of binding" etc.

Reviewer #1

“In their manuscript, the authors describe an interdisciplinary study that characterizes the regulation the Mg²⁺ selective channel MgtE by ATP. To address this question, they provide data from different experimental techniques including X-ray crystallography, titration calorimetry, limited proteolysis, electrophysiology and in vivo studies. The data appears of high quality and the results all support the hypothesis that the binding of ATP shifts the previously described inhibition of the channel by intracellular Mg²⁺ towards physiological concentrations. The same group has previously determined the first structure of the MgtE channel and has characterized its functional properties. This manuscript provides another very important contribution towards the understanding of the structural and functional properties of this important protein family. I thus think that it is an excellent candidate for publication in Nature Communications after some revisions.”

We appreciate Reviewer #1's time and effort, as well as the positive response regarding our manuscript. The specific points mentioned by Reviewer #1 are addressed below.

“I did not have access to the table containing data collection and refinement statistics since it was not included in the files for review. The reported resolution of the full-length channel is 3.6 Å and it is not clear whether this estimate is based on a classical $I/\sigma I$ criterion or by $CC1/2$ that usually increases the nominal high-resolution limit by about 0.2 Å. In any case, the resolution is on the lower side for a detailed atomistic interpretation of the density. This likely creates no problem in this case since the structure at high resolution in complex with Mg²⁺ is known and the described ATP/Mg²⁺ complex adopts a similar conformation. The large ATP ligand can also be identified at this resolution and is supported by the ATP binding observed in the higher resolution structure of the cytoplasmic domains. Still, I think that the authors should carefully document the quality of their structures in the supplemental materials by showing stereo figures of the refined 2Fo-Fc”

We apologize that we forgot to include the table containing the data collection and refinement statistics. We have added the table in the revised manuscript (Table 1). Furthermore, according to this comment, we added a new figure showing the stereo

view of the ATP binding with the $2F_o-F_c$ electron density map (Supplementary Fig. 2).

“The comparably low resolution of the data makes a definite assignment of Mg²⁺ positions ambiguous. Do the authors think that the site for Mg²⁺ 7 shown in Supplementary Fig. 3 is occupied by the ion at all?”

We apologize for our explanation, and agree that the 3.6 Å resolution of the MgtE-ATP complex structure would not be conclusive for the assignment of the Mg²⁺ positions. In particular, the electron density peak for Mg7 is relatively weak (Supplementary Fig. 3). However, the Mg7 site was already assigned and confirmed in the previously reported MgtE structure, by the Co²⁺ anomalous difference Fourier map¹. Therefore, the Mg7 should be real, but considering the electron density peak for Mg7 in the current structure, the Mg7 might be a weak binding site. To avoid any confusion, we added the corresponding description in the revised manuscript (Page 10, lines 8-10).

“Electrophysiology:

· The authors show single channel traces from patches of giant unilamellar liposomes. I believe these represent whole liposome currents and not currents from excised patches. I wonder whether at the describe protein to lipid ratio the authors expect at most one channel per liposomes or whether they also have observed patches containing several channels. In general, it would be interesting to know which fraction of patched liposomes did contain a MgtE channel.”

In our single channel recording, we used excised patches, and did not perform whole cell recording. We apologize for our poor explanation that caused a misunderstanding. We used the recording that showed the typical single channel activity for analysis. We sometimes observed multiple channel openings, and found that ATP still has similar effects on the MgtE activity in such recordings as well. We added the corresponding description in the methods section of the revised manuscript (Page 27, lines 4-6).

”Did the authors ever reconstitute at higher protein to lipid ratio and did they ever observe macroscopic currents with similar behavior.“

We always tested the same protein/lipid ratio, and thus did not see macroscopic currents.

“The authors describe that they have used 5 independent recordings to determine the open probabilities shown in Fig. 2c and 5c,d. What was the duration of each recording and were they from different reconstitutions?”

All of the independent recordings are from different reconstitutions. The durations of the recordings are typically fairly long (at least for more than 5 minutes until the required experiments were done, and then we ended the recordings).

” Which equation was used to fit the curves in Fig. 2c, 5c and d. Is there any indication for cooperativity in the inhibition by Mg²⁺ as evidenced by a Hill slope above 1?”

We used the Hill equation to fit the data. The Hill coefficient values for apo, ATP, and ADP are 0.3, 1.0, and 0.4, respectively. Thus, we did not observe the high cooperativity of the Mg²⁺-dependent inhibition. We added the description of the Hill coefficient values in the revised manuscript (Page 36, Line 18).

“Biochemistry:

· Was Mg²⁺ present in calorimetry experiments, and is there any Mg²⁺ dependence of ATP or ADP binding?”

In the initially submitted manuscript, there was no Mg²⁺ present in the ITC experiments. To answer this question, we conducted new ITC experiments with Mg²⁺. As shown in Figure 1, there is some Mg²⁺ dependency in the ATP binding. With Mg²⁺ ion, MgtE has a higher affinity for ATP. We added these results in the revised manuscript (Page 7, Lines 7-8).

“The mutation of R187 removes a potential Trypsin cleavage site. Did this affect the results of the limited proteolysis experiments?”

We appreciate this comment. Generally speaking, an arginine residue can be a potential

cleavage site for trypsin, but as shown in Supplementary Figure 1, the wild type MgtE was not cleaved at R187, so the mutation at R187 would not affect the results of the limited proteolysis experiments.

General:

“The legends should contain more details of the described figures.”

According to this comment, we add more detailed descriptions in the figure legends (especially for Figures 1 and 4, and Supplementary Figure 2).

“The secondary structure elements in close-up figures are barely visible.”

According to this comment, we enhanced the colors of the secondary structure elements in Figure 3 and Supplementary Figures 2, 3 and 5.

“I do not quite understand the meaning of the brown arrows in Figure 7. I expect that the total influx of Mg^{2+} is dependent on its electrochemical potential and the number and open probabilities of the MgtE channels, given that there is only a single open state with one distinct conductance. If this is the case, why do arrows have a different thickness?”

The brown arrows in Figure 7 were intended to indicate the difference of the open probabilities of MgtE at different Mg^{2+} concentrations, in the presence and absence of ATP (Fig. 2). However, as Reviewer #1 pointed out, the arrows were confusing, and thus we changed the thickness of the brown arrows. In the new Figure 7, all of the brown arrows now have the same thickness, which is less confusing.

Reviewer #2 (Remarks to the Author):

“In this manuscript, Nureki and coworkers present compelling experiments that ATP and Mg are synergistic effectors of channel opening of MgtE. It has long been established that Mg binding the cytosolic domain is a negative regulator of channel

opening, stabilizing the closed conformation of the pump. However, the concentration of Mg required to see these effects in vitro appeared to be too high and not physiological. The authors establish that in the presence of what appears to be physiological ATP concentration, the concentration of Mg required to close the channel is 1-2 mM, or near physiological. The work is novel in that it provides a reasonable clear view of the mechanism of gating by an important metal transporter. The authors show that ATP binds to the CBS domain, and structurally characterize this complex in the presence of bound Mg (up to 7 Mg bind to intact transporter, with at least five in the cytoplasmic domain). Interestingly, in their highest resolution structures, the Mg ion that makes closest approach to ATP is some 5.5 Å away; thus, the authors conclude that ATP and Mg binding are “cooperative” in some way. Mutant transporters were constructed in an effort to validate key features of the ATP binding site in the CBS domain, with most of the work focused on the F227A and R187R mutants. F227 forms a pi-stacking interaction with the purine ring, with R187 makes close approach to the beta- and gamma-phosphate groups of ATP. F227A mutant fails to bind ATP and is nonfunctional; the R187E mutant however, appears to mimic the effects of ATP, via a poorly defined electrostatic model. The work is thorough and the conclusions largely supported by the data; the finding of an accessory role of ATP binding in channel function is not new (see ref. 20), but is interesting nonetheless.”

We appreciate the positive responses and concerns regarding our manuscript. The specific points mentioned by Reviewer #2 are addressed below.

The authors may wish to consider the following comments:

1) The way in which the R187E mutant works requires additional study. The authors seem to imply that the gamma-phosphate of ATP provides an electrostatic environment that screens positive charge thus “attracting” Mg to the cytoplasmic domain to regulate opening. TIn this way the R187E mutant “mimics” the effect of ATP binding. To confirm this, the authors need to measure Mg binding curves in the presence vs. absence of ATP. The expectation is the Mg will bind more tightly to the mutant than to wild-type MgtE. “

We agree that biochemical experiments of Mg^{2+} binding by MgtE and R187 mutants in the presence and absence of ATP by ITC would further strengthen our idea that the negative charges of ATP enhance the affinity of MgtE for Mg^{2+} . This idea is already essentially supported by our electrophysiological analysis of MgtE.

However, the requested ITC experiment is technically impossible for the following reasons.

1. The affinity of MgtE for Mg^{2+} is too weak to measure the binding by ITC. According to the patch clamp analysis, it would be between 5-10 mM, which is beyond the ITC measurement range.
2. Furthermore, the binding of Mg^{2+} ions to MgtE is known to be associated with the multi-step dynamic structural changes of the MgtE cytosolic domain², which would also make the measurement even more difficult and unsuitable for ITC experiments.

Therefore, alternatively, we tested the effect of ADP on the MgtE channel gating by the patch clamp analysis (Fig. 2). ADP contains fewer negatively-charged phosphate groups (ADP:2), as compared to ATP (3). Based on our hypothesis that the negative charges of ATP enhance the affinity of MgtE for Mg^{2+} , ADP would have a weaker effect on the channel gating, and our patch clamp result is indeed consistent with this idea (Fig. 2). We added the corresponding description in the revised manuscript (Page 15, Lines 6-12).

2. “Also, the authors assume that the rate of ATP hydrolysis is slow- was this checked?”

According to this comment, we tested the ATPase activity of MgtE. As shown in Supplementary Figure 7, we did not find significant ATPase activity with the wild type MgtE, as compared with the F227 mutant of MgtE (No ATP binding activity mutant) and apyrase. Accordingly, MgtE either lacks or has very low ATPase activity. This conclusion is also supported by the crystal structures of the MgtE-ATP complex, where we can clearly observe the γ -phosphate group of ATP. We added these descriptions in the revised manuscript (From Page 15, Lines 13 to Page 16, Line 5).

“2) The R187A mutant had the same characteristics of the R187E mutant in a limited proteolysis experiment (Fig. 4). This is incompatible with the speculation in point (1).”

We apologize for our insufficient descriptions that caused the misunderstanding. We agree that the R187A and R187E mutants both had some effect on the limited proteolysis of MgtE, but the intensity of the SDS-PAGE band corresponding to the full length protein for the R187A mutant is weaker than that for the R187E mutant (Fig. 4a). This result is compatible with our idea that the loss of the positive charge and the introduction of the negative charge in the side chain of R187 may have a similar effect to that of ATP binding with negatively-charged phosphate groups. We added these descriptions, as well as the description of the SDS-PAGE band intensity estimation, in the revised manuscript (Page 13, Lines 7-11) (Page 27, Lines 7-12).

“3) The authors need to cite other literature that measures ATP binding affinities to a CBS domain to establish that the binding affinities here are in fact physiological. I would also like to see Mg binding experiments as well, by ITC (see point 1); alternatively the authors could cite the relevant literature.”

According to this comment, we cited other literature that measured ATP binding affinities to a CBS domain (Page 7, Line 10). For the Mg²⁺ binding experiment request, please see our answer above (point 1).

“4) In order to enhance the impact of the work, the authors should revise the Discussion (which is very short and basically recapitulates the Results) and place their findings with MgtE in the context of known mechanisms of metal- or nucleotide-dependent gating in other metal transporter systems. Are there general trends?”

According to this comment, we revised the discussion section to place our findings in the context of the known mechanisms of other channels/transporters, in particular CBS-domain containing proteins and Mg²⁺ channels/transporters (From Page 19, Line 7 to Page 20, Line 15).

“5) The data in Fig. 6 need to be better described. It is unclear why the authors used Co and Ni poisoning as a proxy for Mg transporter function. Why not measure Mg directly in these cells. A better explanation of this experiment may be all that is required. “

We apologize for our poor explanation. Due to the similarity of the coordination chemistry, Mg²⁺ channels/transporters typically permeate Co²⁺ and Ni²⁺. Thus, both the complementation of the Mg²⁺ requirement and the change of the sensitivities to Co²⁺ and Ni²⁺ have been employed to characterize the Mg²⁺ transport system, particularly in bacteria^{3,4}. In fact, the MgtE gene was originally identified based on the restoration of the Mg²⁺ requirement and the Co²⁺ sensitivity⁵. The increased sensitivities to Co²⁺ and Ni²⁺ indicate the “excess” uptake of Co²⁺ and Ni²⁺, as they are toxic, whereas the complementation of the Mg²⁺-requirement shows the “ability” to uptake Mg²⁺.

In the case of MgtE, as shown in Figure 6 as well as in the previous report on MgtE¹, the expression of the cytosolic domain Mg²⁺-binding site mutants leads to the increased sensitivity to Co²⁺ and Ni²⁺, indirectly indicating the loss of the intracellular Mg²⁺-dependent control of Mg²⁺ uptake by MgtE. Similarly, we conducted the Co²⁺ and Ni²⁺ sensitivity assays with the ATP binding site mutants of MgtE, to indirectly evaluate the effects of these mutations on the regulatory function of MgtE for Mg²⁺ homeostasis. We included the corresponding descriptions in the revised manuscript (From Page 16, Line 8 to Page 17, Line 6).

“6) The manuscript requires a thorough editing. For example, p. 13, line 7, “heat of linkage” should be “heat of binding” etc.”

We appreciate this comment. According the comment, with the help of an English proofreading service, we corrected multiple typos in the revised manuscript.

Cover letter References

- 1 Hattori, M. *et al.* Mg²⁺-dependent gating of bacterial MgtE channel underlies Mg²⁺ homeostasis. *The EMBO journal* **28**, 3602-3612, (2009).
- 2 Ishitani, R. *et al.* Mg²⁺-sensing mechanism of Mg²⁺ transporter MgtE probed by molecular dynamics study. *Proceedings of the National Academy of Sciences of*

- the United States of America* **105**, 15393-15398, (2008).
- 3 Moomaw, A. S. & Maguire, M. E. The unique nature of mg^{2+} channels. *Physiology* **23**, 275-285, (2008).
- 4 Maguire, M. E. & Cowan, J. A. Magnesium chemistry and biochemistry. *Biometals : an international journal on the role of metal ions in biology, biochemistry, and medicine* **15**, 203-210 (2002).
- 5 Smith, R. L., Thompson, L. J. & Maguire, M. E. Cloning and characterization of MgtE, a putative new class of Mg^{2+} transporter from *Bacillus firmus* OF4. *Journal of bacteriology* **177**, 1233-1238 (1995).

Reviewers' Comments:

Reviewer #1:

Remarks to the Author:

I have read the revision and think that the authors have addressed my comments in a satisfactory manner.

Reviewer #2:

Remarks to the Author:

In this revised manuscript, the authors have thoughtfully responded to most of the comments that were raised in the previous review, with the exception of the characterization of the R187E mutant which still strikes this reviewer as not so compelling. The new discussion is markedly improved. The authors may wish to consider the following comments.

Major:

1) Fig, 4b: The authors attempt Mg binding to R187E MgtE (in the presence of ATP) as suggested in my previous comments and claim that on the basis of this experiment, R187E binds three-fold less tightly than wild-type (p. 14, lines 207-209). This is not a publication-quality binding curve and little if any quantitative information can be obtained from this experiment. I suggest the authors repeat it at higher protein concentration to obtain higher heats or remove the data altogether. Also, does this mutant show any Mg association in the absence of ATP (suggested previously)? This experiment should also be tried.

Minor:

1) Mg binding site ligands (Fig. S6): Many of the Mg binding site ligands are apparently not conserved (Mg5, Mg6) or the arrows reflect coordination by backbone atoms. I suggest that the authors highlight side chain vs. main chain "coordination" in the figure legend, at least, so as not to confuse the reader.

2) Legend, Fig. S5: Add the corresponding residues in the CLC channel that correspond to the conserved residues the authors are highlighting in the figure.

3) Discussion (lines 307-311): Define CNNM (CBS domain divalent metal cation transport mediator) for the general reader. Also, the authors may wish to cite recent structural and functional work (see Tremblay and collaborators; Gimenez-Mascareli et al., JBC 2017) to suggest that CNNM transporters are thought to be effluxers. Thus, Mg or Mg-ATP might be expected to function as an activator in CNNM transporters. This should be briefly discussed in the context of MgtE so as to further inform the reader.

Reviewer #1

“I have read the revision and think that the authors have addressed my comments in a satisfactory manner.”

Thank you so much for the time you spent reviewing our manuscript.

Reviewer #2 (Remarks to the Author):

“In this revised manuscript, the authors have thoughtfully responded to most of the comments that were raised in the previous review, with the exception of the characterization of the R187E mutant which still strikes this reviewer as not so compelling. The new discussion is markedly improved. The authors may wish to consider the following comments.”

We appreciate the positive responses regarding our revised manuscript. The comments mentioned by Reviewer #2 are addressed below.

Major:

1) Fig, 4b: The authors attempt Mg binding to R187E MgtE (in the presence of ATP) as suggested in my previous comments and claim that on the basis of this experiment, R187E binds three-fold less tightly than wild-type (p. 14, lines 207-209). This is not a publication-quality binding curve and little if any quantitative information can be obtained from this experiment. I suggest the authors repeat it at higher protein concentration to obtain higher heats or remove the data altogether. Also, does this mutant show any Mg association in the absence of ATP (suggested previously)? This experiment should also be tried.

Reviewer #2 may have a misunderstanding regarding our previous revised manuscript, and we would like to explain.

First, Reviewer #2 may have misunderstood that we conducted the biochemical experiments of Mg²⁺ binding with the R187 mutant by ITC in the revised manuscript (Fig. 4b). However, Fig. 4b was already included in our initially-submitted manuscript,

and shows the ATP binding of the R187E mutant, not the Mg^{2+} binding. We apologize for the misunderstanding we caused.

As we explained in the cover letter of the previous revised manuscript, the requested ITC experiment is technically impossible, and thus as an alternative, we tested the effect of ADP on the MgtE channel gating by the patch clamp analysis (Fig. 2). For the details, please refer the following sentences below (shown in italics). These are pasted from our previous cover letter.

“We agree that biochemical experiments of Mg^{2+} binding by MgtE and R187 mutants in the presence and absence of ATP by ITC would further strengthen our idea that the negative charges of ATP enhance the affinity of MgtE for Mg^{2+} . This idea is already essentially supported by our electrophysiological analysis of MgtE.

However, the requested ITC experiment is technically impossible for the following reasons.

- 1. The affinity of MgtE for Mg^{2+} is too weak to measure the binding by ITC. According to the patch clamp analysis, it would be between 5-10 mM, which is beyond the ITC measurement range.*
- 2. Furthermore, the binding of Mg^{2+} ions to MgtE is known to be associated with the multi-step dynamic structural changes of the MgtE cytosolic domain¹, which would also make the measurement even more difficult and unsuitable for ITC experiments.*

Therefore, alternatively, we tested the effect of ADP on the MgtE channel gating by the patch clamp analysis (Fig. 2). ADP contains fewer negatively-charged phosphate groups (ADP:2), as compared to ATP (3). Based on our hypothesis that the negative charges of ATP enhance the affinity of MgtE for Mg^{2+} , ADP would have a weaker effect on the channel gating, and our patch clamp result is indeed consistent with this idea (Fig. 2). We added the corresponding description in the revised manuscript (Page 15, Lines 6-12).”

We hope this will solve the misunderstanding. As for the concern regarding the quality of the binding curve in Figure 4b, we agree that the quality of the ATP binding curve from the R187E mutant is not as good as that of the wild type, due to the weaker affinity,

and ideally it would be better to obtain a clearer curve at a higher protein concentration. However, this is technically impossible for the following reasons.

1. The higher protein concentration needs to be the concentration of the gel filtration fractions after purification.
2. However, concentrating the protein will also increase the DDM detergent concentration, and the increased DDM concentration causes a big increase in the heat noise. This problem can be serious for measurements of weak binding, including the interaction between MgtE and ATP.
3. Therefore, we directly used the gel filtration sample without concentration, and thus the current tested protein concentration (3-4 mg/ml) is the maximum protein concentration we could try.

Nevertheless, as Reviewer #2 pointed out, we agree that although the current ITC data on the R187E mutant may not be quantitative, the data still qualitatively show the weaker binding to ATP, as compared to the binding of the wild type to ATP. Therefore, in the revised manuscript, we removed the K_d value of the R187E mutant for ATP, and toned down the phrase regarding the ITC data of the R187E mutant (Page 14, Lines 1-3).

Minor:

1) Mg binding site ligands (Fig. S6): Many of the Mg binding site ligands are apparently not conserved (Mg5, Mg6) or the arrows reflect coordination by backbone atoms. I suggest that the authors highlight side chain vs. main chain “coordination” in the figure legend, at least, so as not to confuse the reader.

Thank you so much for the comment. According to the comment, we highlighted the side chain vs. main chain “coordination” in the figure legend of Supplementary Figure S6.

2) Legend, Fig. S5: Add the corresponding residues in the CLC channel that correspond to the conserved residues the authors are highlighting in the figure.

According to this comment, we added the residues in the CLC channel that correspond to the conserved residues we highlighted in the figure legend of Supplementary Figure S5.

3) Discussion (lines 307-311): Define CNNM (CBS domain divalent metal cation transport mediator) for the general reader. Also, the authors may wish to cite recent structural and functional work (see Tremblay and collaborators; Gimenez-Mascareli et al., JBC 2017) to suggest that CNNM transporters are thought to be effluxers. Thus, Mg or Mg-ATP might be expected to function as an activator in CNNM transporters. This should be briefly discussed in the context of MgtE so as to further inform the reader.

According to this comment, we cited the recent structural and functional work on CNNM transporters, and added a discussion about them in the context of MgtE (From Page 20, Line 10 to Page 21, Line 2).

Cover letter References

- 1 Ishitani, R. *et al.* Mg²⁺-sensing mechanism of Mg²⁺ transporter MgtE probed by molecular dynamics study. *Proceedings of the National Academy of Sciences of the United States of America* **105**, 15393-15398, (2008).